# Enantioselective Synthesis of the Active Sex Pheromone Components of the Female Lichen Moth, *Lyclene dharma dharma*, and Their Enantiomers

**DOI:** 10.3390/molecules29122918

**Published:** 2024-06-19

**Authors:** Yun Zhou, Jianan Wang, Yueru Zhang, Xiaochen Fu, Hongqing Xie, Jinlong Han, Jianhua Zhang, Jiangchun Zhong, Chenggang Shan

**Affiliations:** 1Institute of Industrial Crops, Shandong Academy of Agricultural Sciences, Jinan 250100, China; zysass2021@163.com (Y.Z.); 13553169667@163.com (Y.Z.); fxc13345203071@163.com (X.F.); xiehongqing@zju.edu.cn (H.X.); goldendragonh@163.com (J.H.); zhangjianhua198904@163.com (J.Z.); 2Department of Applied Chemistry, China Agricultural University, Beijing 100193, China; xxjnwang@163.com (J.W.); zhong@cau.edu.cn (J.Z.)

**Keywords:** sex pheromone, *Lyclene dharma dharma*, Evans’ chiral auxiliary, enantioselective synthesis

## Abstract

The Lichen moth, *Lyclene dharma dharma* (*Arctiidae*, *Lithosiinae*), plays a significant role in forest ecosystem dynamics. A concise and novel method to synthesize the active sex pheromone components, (*S*)-14-methyloctadecan-2-one ((*S*)-**1**), (*S*)-6-methyloctadecan-2-one ((*S*)-**2**), and their enantiomers has been developed. Key steps in the synthesis include the use of Evans’ chiral auxiliaries, Grignard cross-coupling reactions, hydroboration–oxidation, and Wacker oxidation. The synthesized sex pheromone components hold potential value for studies on communication mechanisms, species identification, and ecological management.

## 1. Introduction

The Lichen moth, *Lyclene dharma dharma* (*Arctiidae*, *Lithosiinae*), is a species within the Lithosiinae subfamily of the Arctiidae family. It is notable as the first species in this subfamily for which the sex pheromone has been identified [1]. This moth is distributed globally, and its larvae predominantly feed on forest lichen, playing a crucial role in supporting the forest ecosystem [2]. In 2007, Ando and his coworkers identified three electroantennographically active components in female moths of *Lyclene dharma dharma* (*Arctiidae*, *Lithosiinae*), namely 14-methyl-2-octadecanone, 6-methyl-2-octadecanone, and 6,14-dimethyl-2-octadecanone; they synthesized these racemic sex pheromone components and found them in a ratio of approximately 1:2:1 [1,2,3]. Subsequently, they confirmed that (*S*)-14-methyloctadecan-2-one ((*S*)-**1**) and (*S*)-6-methyloctadecan-2-one ((*S*)-**2**) are active pheromone components (Figure 1) and 6,14-dimethyl-2-octadecanone was not indispensable for bioactivity [4].

The synthesis of the active sex pheromone components (*S*)-**1**, (*S*)-**2**, and their enantiomers ((*R*)-**1**, (*R*)-**2**) was previously based on a chiral pool strategy. In 2009, Mori completed the synthesis of sex pheromone components in *Lyclene dharma dharma*, the synthesis of (*S*)-**1** with a total yield of 6% in 13 steps, (*S*)-**2** with a total yield of 20% in 11 steps, using enantiomers of citronellal and methyl (*S*)-3-hydroxy-2-methylpropanoate as chiral starting materials, with olefin cross metathesis as the key reaction [5]. They then employed the Wittig reaction, alkylation of alkynes, and acetoacetic ester synthesis to construct sex pheromone components in *Lyclene dharma dharma* from (*S*)- and (*R*)-citronellal; they synthesized (*S*)-**1** with a total yield of 11% in 15 steps, (*S*)-**2** with a total yield of 10% in 13 steps [6]. In 2023, Kobayashi and colleagues reported the synthesis of (*S*)-14-methyloctadecan-2-one with a total yield of 10% in seven steps, using (*S*)-3-butyn-2-ol as a chiral starting material, with a copper-catalyzed coupling reaction of alkynyl 2-pyridyl sulfonate and an alkyl lithium reagent as the key reaction [7]. The reported synthetic routes still have some shortcomings, such as difficult synthetic routes, expensive chiral starting materials, complex synthetic operations, and poor overall yields. To facilitate future applications of the active sex pheromone components in exploring the communication mechanisms of Lichen moths, species-specific studies and ecological management, and developing a concise and efficient synthetic method, are still in demand [8]. Herein, we synthesized the active sex pheromone components of *Lyclene dharma dharma* (*Arctiidae*, *Lithosiinae*) and their enantiomers by an enantioselective synthesis strategy. Our methodology employed Evans’ chiral auxiliaries, a Grignard cross-coupling reaction, hydroboration–oxidation and Wacker oxidation as key reactions, enabled straightforward synthesis, and yielded the target pheromone with high enantiomeric purity.

## 2. Results and Discussion

### 2.1. Retrosynthetic Analysis

The retrosynthetic analysis of (*S*)-14-methyloctadecan-2-one ((*S*)-**1**) is delineated in Figure 1. The target sex pheromone (*S*)-**1** could be synthesized via Wacker oxidation of alkyl chiral terminal alkene (*S*)-**12**. The coupling of but-3-en-1-ylmagnesium bromide **11** with the sulfonate ester of chiral primary alcohol (*S*)-**10** yielded (*S*)-**12**. Chiral primary alcohol (*S*)-**10** was assembled from Grignard cross-coupling and hydroboration–oxidation of chiral enol (*R*)-**7**, which itself is derived from the reduction of the oxazolidinone amide. Most critically, the chiral methyl of (*R*, *R*)-**6** could be introduced by Evans’ chiral auxiliaries after the acylation of undec-10-enoic acid **3** with (*R*)-4-isopropyloxazolidin-2-one ((*R*)-**4**), followed by diastereoselective methylation of the oxazolidinone amide.

### 2.2. Synthesis of Chiral Primary Alcohol ***7***

As illustrated in Figure 2, our synthetic sequence was initiated with the reaction of undec-10-enoic acid (**3**) with pivaloyl chloride, which was then converted to oxazolidinone amide (*R*)-**5** in 92% yield through acylation of (*R*)-4-isopropyloxazolidin-2-one ((*R*)-**4**) [9,10]. Enantioselective methylation was then carried out, using methyl iodide (MeI) in the presence of sodium bis(trimethylsilyl)amide (NaHMDS), resulting in (*R*)-4-isopropyl-3-((*R*)-2-methylundec-10-enoyl) oxazolidin-2-one ((*R*, *R*)-**6**) [11,12]. This step achieved an 82% yield. Following the established protocols of Evans’ chiral auxiliary, the newly formed stereocenter at C2 of (*R*, *R*)-**6** favored the *R* configuration [13]. Reductive cleavage of the chiral auxiliary with lithium aluminum hydride (LiAH_4_) afforded (*R*)-2-methylundec-10-en-1-ol ((*R*)-**7**) in 88% yield and 99% ee [14]. The enantiomeric excess of (*R*)-**7** was estimated by ^1^H NMR spectra of its corresponding Mosher ester derivative [15]. The specific rotation of (*R*)-**7** was +12.3 (c 1.17, CHCl_3_), which matched the reported value in the literature of +8.4 (c 4.6, CHCl_3_), which also supported that the newly constructed chiral methyl configuration is the *R* configuration [16]. In the same manner, (*S*)-2-methylundec-10-en-1-ol ((*S*)-**7**) was synthesized from undec-10-enoic acid (**3**) and (*S*)-4-benzyloxazolidin-2-one ((*S*)-**4**) through a series of steps including acylation, enantioselective methylation, and reduction with slightly different yields.

### 2.3. Synthesis of Chiral Long Chain Terminal Alkene ***12***

The construction of the chiral long chain terminal alkene **12** is shown in Figure 3. (*S*)-2-Methylundec-10-en-1-ol ((*S*)-**7**) was activated with tosyl chloride (TsCl) to yield a *p*-toluenesulfonyl intermediate, which coupled with propylmagnesium bromide and catalytic lithium tetrachlorocuprate (II) (Li_2_CuCl_4_) to afford (*S*)-10-methyltetradec-1-ene ((*S*)-**9**) in 83% yield [17]. The hydroboration–oxidation of (*S*)-**9** by 9-borabicyclo [3.3.1] nonane (9-BBN) and hydrogen peroxide (H_2_O_2_) yielded (*S*)-10-methyltetradecan-1-ol ((*S*)-**10**) with a 90% yield [18]. Subsequent tosylation of compound (*S*)-**10** with TsCl, followed by coupling reaction with but-3-en-1-ylmagnesium bromide, smoothly produced (*S*)-14-methyloctadec-1-ene ((*S*)-**12**) in 80% yield [19,20]. According to similar steps, (*R*)-14-methyloctadec-1-ene ((*R*)-**12**) was constructed from (*R*)-2-methylundec-10-en-1-ol ((*R*)-**7**).

### 2.4. Synthesis of the Active Components of Sex Pheromone ***1***

With the necessary chiral terminal alkenes **12** prepared, we directed our efforts toward synthesizing the target pheromone (*S*)-**1** and its enantiomer (*R*)-**1**, as outlined in Figure 4. Wacker oxidation of (*S*)-14-methyloctadec-1-ene ((*S*)-**12**) using PdCl_2_, CuCl_2_, and DMF/H_2_O (7:1) conditions yielded the target compound (*S*)-14-methyloctadecan-2-one ((*S*)-**1**) in 73% yield [21,22]. The specific rotation of (*S*)-**1** was + 0.92 (c 0.43, CHCl_3_), aligning with the reported value in the literature of +1.05 (c 3.13, hexane) [5]. Similarly, (*R*)-14-methyloctadecan-2-one ((*R*)-**1**) was synthesized from (*R*)-14-methyloctadec-1-ene ((*R*)-**12**) using a comparable Wacker oxidation process in 70% yield. The specific rotation of (*R*)-**1** was −0.89 (c 0.43, CHCl_3_), aligning with the reported value in the literature of −0.6 (c 3.07, hexane). Additionally, the structures of (*S*)-**1** and (*R*)-**1** were confirmed using ^1^H NMR, ^13^C NMR (Appendix A) and HRMS spectra, consistent with the literature [5].

### 2.5. Synthesis of Chiral Primary Alcohol ***16***

The synthetic route for sex pheromone (*S*)-**2** was analogous to that for (*S*)-**1**. As illustrated in Figure 5, we synthesized (*S*)-4-isopropyl-3-tetradecanoyloxazolidin-2-one ((*S*)-**14**) by reacting tetradecanoic acid (**13**) with the (*S*)-Evans’ chiral auxiliary ((*S*)-**4**) in 81% yield [9,10]. Subsequent asymmetric methylation using sodium bis(trimethylsilyl)amide (NaHMDS) and methyl iodide (MeI) produced (*S*)-4-isopropyl-3-((*S*)-2-methyltetradecanoyl) oxazolidin-2-one ((*S*)-**15**) in 75% yield [12]. Reduction of compound (*S*)-**15** with lithium aluminum hydride (LiAH_4_) yielded (*S*)-2-methyltetradecan-1-ol ((*S*)-**16**) in 77% yield [14], with a specific rotation of −10.59 (c 1.13, CHCl_3_), consistent with the value reported in the literature of −11.36 (c 0.88, CHCl_3_) [23]. Similarly, (*R*)-2-methyltetradecan-1-ol ((*R*)-**16**) was synthesized using identical steps starting with tetradecanoic acid (**13**) and (*R*)-Evans’ chiral auxiliary ((*R*)-**4**), achieving a specific rotation of +6.5 (c 1.35, CHCl_3_), which compared favorably with the reported value of +4.5 (c 1.68, CHCl_3_) [23].

### 2.6. Synthesis of Bioactive Components of Sex Pheromone ***2***

Following the successful synthesis of the key intermediate (*S*)-**16**, we aimed to synthesize the target pheromone (*S*)-**2** and its enantiomer (*R*)-**2**, as shown in Figure 6. Compound (*S*)-**16** was converted into the activated sulfonate ester through reaction with *p*-toluenesulfonyl chloride (TsCl), followed by coupling with but-3-en-1-ylmagnesium bromide under the catalysis of lithium tetrachlorocuprate (II) (Li_2_CuCl_4_) affording (*S*)-6-methyloctadec-1-ene ((*S*)-**17**) in 71% yield [20]. The final transformation into the target pheromone (*S*)-6-methyloctadecan-2-one ((*S*)-**2**) was accomplished through Wacker oxidation of (S)-**17** [21,22], where its specific rotation of −0.93 (c 1.72, CHCl_3_) was consistent with a reported value of −0.38 (c 3.63, hexane) [5]. Similarly, the enantiomer (*R*)-6-methyloctadecan-2-one ((*R*)-**2**) was synthesized along a parallel pathway, exhibiting a specific rotation of +0.88 (c 0.45, CHCl_3_), in line with a reported value of +0.35 (c 4.08, hexane). Additionally, the spectroscopic data {^1^H NMR, ^13^C NMR (Appendix A) and HRMS} for both (*S*)-**2** and (*R*)-**2** corresponded with the literature [5].

## 3. Materials and Methods

### 3.1. General Information

All reactions were conducted under an inert argon atmosphere within a Schlenk line system. Commercially available reagents were used as received, while solvents were distilled prior to use according to standard procedures. ^1^HNMR spectra (500 MHz, TMS at δ 0.00 ppm or CDCl_3_ at δ 7.26 ppm) and ^13^CNMR spectra (126 MHz, CDCl_3_ at δ 77.16 ppm as internal standard) were recorded on a Bruker DP-X500 spectrometer (Bruker Corporation, Beijing, China). High-resolution mass spectra (HRMS) were recorded on a Waters LCT Premier™ (Waters Corporation, Beijing, China) equipped with an electrospray ionization (ESI) mass spectrometer. Optical rotations were recorded on a Rudolph Research Analytical AUTOPOL-IV polarimeter (Rudolph Research Analytical, Beijing, China). Melting points were measured by a Stuart SMP3 Melt-Temp apparatus (Stuart Equipment, Beijing, China) and are reported uncorrected.

### 3.2. Synthesis of (R)-4-Isopropyl-3-(undec-10-enoyl)oxazolidin-2-one ((R)-***5***)

Under an argon atmosphere, triethylamine (1.66 mL, 12.0 mmol, 2.0 equiv.) and pivaloyl chloride (0.90 mL, 7.2 mmol, 1.2 equiv.) were added to a stirred solution of undec-10-enoic acid (**3**) (1.11 g, 6.0 mmol, 1.0 equiv.) in dry THF (30 mL) at −78 °C sequentially. The reaction mixture was then stirred for 30 min before being allowed to warm to room temperature. After stirring for an additional hour at room temperature, lithium chloride (0.76 g, 18.0 mmol, 3.0 equiv.) was introduced. The mixture was re-cooled to −78 °C, and a solution of (*R*)-4-benzyl-2-oxazolidinone (*R*)-**4** (0.93 g, 7.2 mmol, 1.2 equiv.) in dry THF (10 mL) was added slowly. Stirring was continued for one hour at −78 °C, after which the mixture was allowed to warm to room temperature slowly and stirred for another 12 h. The reaction was quenched by adding water (20 mL) and extracted with ethyl acetate (3 × 50 mL). The two phases were separated, and the organic layer was washed with brine (100 mL). The combined organic extracts were dried over anhydrous sodium sulfate and concentrated under reduced pressure. Purification by silica gel column chromatography using a mobile phase of ethyl acetate and petroleum ether (1:5) yielded (*R*)-4-isopropyl-3-(undec-10-enoyl)oxazolidin-2-one (*R*)-**5** (1.63 g, 92% yield) as a colorless oil. [α]_D_^25^ = −85.7 (c 2.21, CHCl_3_). ^1^H NMR (500 MHz, CDCl_3_) δ 5.79 (ddt, *J* = 16.9, 10.1, 6.7 Hz, 1H), 4.97 (d, *J* = 17.1 Hz, 1H), 4.91 (d, *J* = 10.2 Hz, 1H), 4.42 (dt, *J* = 7.2, 3.3 Hz, 1H), 4.25 (t, *J* = 8.7 Hz, 1H), 4.19 (dd, *J* = 9.1, 2.8 Hz, 1H), 2.96 (dd, *J* = 16.1, 7.6 Hz, 1H), 2.83 (dt, *J* = 16.1, 7.5 Hz, 1H), 2.35 (ddd, *J* = 13.8, 8.9, 5.5 Hz, 1H), 2.02 (q, *J* = 7.0 Hz, 2H), 1.64 (dq, *J* = 13.0, 6.8 Hz, 2H), 1.37–1.28 (m, 10H), 0.90 (d, *J* = 7.0 Hz, 3H), 0.86 (d, *J* = 6.9 Hz, 3H). ^13^C NMR (126 MHz, CDCl_3_) δ 173.53, 154.18, 139.30, 114.23, 63.41, 58.48, 35.62, 33.89, 29.41, 29.20, 29.16, 29.00, 28.49, 24.55, 18.07, 14.76. HRMS (ESI, *m*/*z*): calculated for [M + H]^+^ C_17_H_30_NO_3_ 296.2220, found: 296.2214.

### 3.3. Synthesis of (R)-4-Isopropyl-3-((R)-2-methylundec-10-enoyl)oxazolidin-2-one ((R, R)-***6***)

Under an argon atmosphere, to a stirred solution of (*R*)-4-isopropyl-3-(undec-10-enoyl)oxazolidin-2-one (*R*)-**5** (1.57 g, 4.6 mmol, 1.0 equiv.) in dry tetrahydrofuran (10 mL) was treated with sodium bis(trimethylsilyl)amide (4.60 mL, 2.0 M in THF, 9.2 mmol, 2.0 equiv.) via a syringe pump over 30 min at −78 °C. The resulting mixture was stirred for 1 h at −78 °C, followed by the slow addition of methyl iodide (1.43 mL, 23.0 mmol, 5.0 equiv.). The reaction mixture was maintained for 2 h at −78 °C, then allowed to warm to −50 °C and stirred for an additional 6 h. The reaction was quenched with saturated aqueous ammonium chloride (20 mL). After phase separation, the aqueous layer was extracted three times with ethyl acetate (3 × 50 mL). The ethyl acetate extracts were combined with the organic layer, washed with saturated aqueous sodium chloride (100 mL), dried over anhydrous sodium sulfate, and filtered. The filtrate was concentrated under reduced pressure. Purification by silica gel column chromatography using ethyl acetate/petroleum ether (1:10) yielded (*R*)-4-isopropyl-3-((*R*)-2-methylundec-10-enoyl)oxazolidin-2-one ((*R*, *R*)-**6**) as a light yellow oil (1.17 g, 82% yield). [α]_D_^25^ = −88.68 (c 0.91, CHCl_3_). ^1^H NMR (500 MHz, CDCl_3_) δ 5.80 (ddt, *J* = 16.9, 10.1, 6.7 Hz, 1H), 4.98 (d, *J* = 16.9 Hz, 1H), 4.92 (d, *J* = 10.2 Hz, 1H), 4.44 (dt, *J* = 7.4, 3.3 Hz, 1H), 4.26 (t, *J* = 8.7 Hz, 1H), 4.19 (dd, *J* = 9.1, 2.7 Hz, 1H), 3.72 (h, *J* = 6.8 Hz, 1H), 2.34 (dp, *J* = 10.9, 4.1 Hz, 1H), 2.02 (q, *J* = 7.0 Hz, 2H), 1.73–1.69 (m, 1H), 1.36-1.26 (m, 11H), 1.19 (d, *J* = 6.9 Hz, 3H), 0.91 (d, *J* = 7.0 Hz, 3H), 0.87 (d, *J* = 6.9 Hz, 3H). ^13^C NMR (126 MHz, CDCl_3_) δ 177.45, 153.81, 139.34, 114.25, 63.32, 58.57, 37.85, 33.91, 33.23, 29.73, 29.46, 29.19, 29.03, 28.56, 27.41, 18.08, 18.00, 14.82. HRMS (ESI, *m*/*z*): calculated for [M + H]^+^ C_18_H_32_NO_3_ 310.2377, found: 310.2369.

### 3.4. Synthesis of (R)-2-Methylundec-10-en-1-ol ((R)-***7***)

Under an argon atmosphere, to a stirred solution of LiAlH_4_ (0.43 g, 11.2 mmol, 3.5 equiv.) in dry tetrahydrofuran (10 mL) was added (*R*)-4-isopropyl-3-((*R*)-2-methylundec-10-enoyl) oxazolidin-2-one ((*R*, *R*)-***6***) (0.99 g, 3.2 mmol, 1.0 equiv.) in dry tetrahydrofuran (6 mL) at 0 °C. The resulting mixture was allowed to warm to room temperature and stirred for 12 h. The reaction was quenched with saturated aqueous ammonium chloride (10 mL) and diluted with ethyl acetate. The precipitate was filtered off, and the filtrate was then dried over anhydrous sodium sulfate and filtered. The filtrate was concentrated under reduced pressure. Purification by silica gel column chromatography using ethyl acetate/petroleum ether (1:5) yielded (*R*)-2-methylundec-10-en-1-ol ((*R*)-**7**) (0.52 g, 88% yield, ≥99% ee, determined by ^1^H NMR analysis of the ester derived from (*S*)-MTPACl) as a colorless oil. [α]_D_^25^ = +12.3 (c 1.17, CHCl_3_). ^1^H NMR (500 MHz, CDCl_3_) δ 5.81 (ddt, *J* = 16.9, 10.2, 6.7 Hz, 1H), 4.99 (d, *J* = 17.1 Hz, 1H), 4.92 (d, *J* = 10.2 Hz, 1H), 3.50 (dd, *J* = 10.5, 5.8 Hz, 1H), 3.41 (dd, *J* = 10.4, 6.6 Hz, 1H), 2.03 (q, *J* = 7.0 Hz, 2H), 1.63–1.57 (m, 1H), 1.41–1.28 (m, 12H), 1.12-1.06 (m, 1H), 0.91 (d, *J* = 6.7 Hz, 3H). ^13^C NMR (126 MHz, CDCl_3_) δ 139.36, 114.25, 68.55, 35.89, 33.94, 33.27, 30.01, 29.60, 29.26, 29.06, 27.10, 16.72. HRMS (ESI, *m*/*z*): calculated for [M + H]^+^ C_12_H_25_O 185.1900, found: 185.1898.

### 3.5. Synthesis of (S)-4-Isopropyl-3-(undec-10-enoyl)oxazolidin-2-one((S)-***5***)

Following the procedure previously described for the synthesis of (*R*)-**5**, undec-10-enoic acid (**3**) (1.11 g, 6.0 mmol, 1.0 equiv.) and (*S*)-4-benzyl-2-oxazolidinone (*S*)-**4** (0.93 g, 7.2 mmol, 1.2 equiv.) were reacted to yield (*S*)-4-isopropyl-3-(undec-10-enoyl)oxazolidin-2-one((*S*)-**5**) (1.54 g, 87% yield) as a colorless oil. [α]_D_^25^ = +74.16 (c 3.33, CHCl_3_) ^1^H NMR (500 MHz, CDCl_3_) δ 5.78 (ddt, *J* = 16.9, 10.1, 6.7 Hz, 1H), 4.96 (d, *J* = 17.1 Hz, 1H), 4.90 (d, *J* = 10.2 Hz, 1H), 4.41 (dt, *J* = 7.3, 3.4 Hz, 1H), 4.24 (t, *J* = 8.7 Hz, 1H), 4.18 (dd, *J* = 9.1, 2.9 Hz, 1H), 2.99–2.92 (m, 1H), 2.86–2.79 (m, 1H), 2.38–2.32 (m, 1H), 2.01 (q, *J* = 7.0 Hz, 2H), 1.69–1.57 (m, 2H), 1.36–1.27 (m, 10H), 0.89 (d, *J* = 7.1 Hz, 3H), 0.85 (d, *J* = 6.9 Hz, 3H). ^13^C NMR (126 MHz, CDCl_3_) δ 173.47, 154.13, 139.23, 114.19, 63.37, 58.43, 35.57, 33.85, 29.36, 29.16, 29.12, 28.95, 28.45, 24.51, 18.02, 14.72. HRMS (ESI, *m*/*z*): calculated for [M + Na]^+^ C_17_H_29_NO_3_Na 318.2040, found: 318.2034.

### 3.6. Synthesis of (S)-4-Isopropyl-3-((S)-2-methylundec-10-enoyl)oxazolidin-2-one ((S, S)-***6***)

Following the procedure previously described for the synthesis of (*R*, *R*)-**6**, (*S*)-4-isopropyl-3-(undec-10-enoyl) oxazolidin-2-one (*S*)-**5** (1.18 g, 4.0 mmol, 1.0 equiv.) and methyl iodide (1.25 mL, 20.0 mmol, 5.0 equiv.) were reacted to yield (*S*)-4-isopropyl-3-((*S*)-2-methylundec-10-enoyl) oxazolidin-2-one ((*S*, *S*)-**6**) (1.05 g, 85% yield) as a colorless oil. [α]_D_^25^ = +75.59 (c 0.85, CHCl_3_). ^1^H NMR (500 MHz, CDCl_3_) δ 5.74 (ddt, *J* = 16.9, 10.2, 6.7 Hz, 1H), 4.92 (dq, *J* = 17.1, 1.6 Hz, 1H), 4.86 (ddd, *J* = 10.2, 2.1, 1.0 Hz, 1H), 4.40–4.36 (m, 1H), 4.19 (t, *J* = 8.7 Hz, 1H), 4.13 (dd, *J* = 9.1, 3.0 Hz, 1H), 3.69–3.62 (m, 1H), 2.32–2.25 (m, 1H), 1.99–1.94 (m, 2H), 1.67–1.61 (m, 1H), 1.33–1.26 (m, 3H), 1.23–1.18 (m, 8H), 1.13 (d, *J* = 6.9 Hz, 3H), 0.84 (d, *J* = 7.0 Hz, 3H), 0.81 (d, *J* = 6.9 Hz, 3H). ^13^C NMR (126 MHz, CDCl_3_) δ 177.46, 153.82, 139.36, 114.26, 63.33, 58.58, 37.87, 33.92, 33.25, 29.74, 29.47, 29.20, 29.04, 28.58, 27.43, 18.09, 18.01, 14.83. HRMS (ESI, *m*/*z*): calculated for [M + Na]^+^ C_18_H_31_NO_3_ Na 332.2196, found: 332.2193.

### 3.7. Synthesis of (S)-2-Methylundec-10-en-1-ol ((S)-***7***)

Following the procedure previously described for the synthesis of (*R*)-**7**, (*S*)-4-isopropyl-3-((*S*)-2-methylundec-10-enoyl) oxazolidin-2-one ((*S*, *S*)-**6**) (0.93 g, 3.0 mmol, 1.0 equiv.) and LiAlH_4_ (0.40 g, 10.5 mmol, 3.5 equiv.) were reacted to yield (*S*)-2-methylundec-10-en-1-ol ((*S*)-**7**) (0.45 g, 82% yield, ≥99% ee, determined by ^1^H NMR analysis of the ester derived from (*S*)-MTPACl) as a colorless oil. [α]_D_^25^ = −8.70 (c 0.87, CHCl_3_). ^1^H NMR (500 MHz, Chloroform-*d*) δ 5.81 (ddt, *J* = 16.9, 10.2, 6.7 Hz, 1H), 4.99 (dq, *J* = 17.1, 1.6 Hz, 1H), 4.93 (ddt, *J* = 10.2, 2.2, 1.2 Hz, 1H), 3.50 (dd, *J* = 10.5, 5.8 Hz, 1H), 3.41 (dd, *J* = 10.5, 6.5 Hz, 1H), 2.06–2.01 (m, 2H), 1.65–1.56 (m, 1H), 1.41–1.36 (m, 4H), 1.33–1.27 (m, 8H), 1.13–1.07 (m, 1H), 0.91 (d, *J* = 6.7 Hz, 3H). ^13^C NMR (126 MHz, CDCl_3_) δ 139.37, 114.26, 68.57, 35.91, 33.94, 33.28, 30.02, 29.61, 29.26, 29.07, 27.10, 16.73. HRMS (ESI, *m*/*z*): calculated for [M + H]^+^ C_12_H_25_O 185.1900, found: 185.1901

### 3.8. Synthesis of (S)-10-Methyltetradec-1-ene ((S)-***9***)

Under an argon atmosphere, to a solution of (*R*)-2-methylundec-10-en-1-ol ((*R*)-**7**) (0.46 g, 2.5 mmol, 1.0 equiv.) and triethylamine (0.61 mL, 4.4 mmol, 1.75 equiv.) in dichloromethane (6 mL) was added tosyl chloride (0.95 g, 5.0 mmol, 2.0 equiv.) in dichloromethane (10 mL) slowly. The mixture was stirred for 24 h at rt. Afterward, the reaction mixture was quenched with saturated aqueous sodium bicarbonate (20 mL), extracted with dichloromethane (3 × 30 mL), washed with brine, and dried over anhydrous sodium sulfate. Following concentration, the product was purified using column chromatography on silica gel with ethyl acetate/petroleum ether (1:5) to yield the desired tosylate product as a colorless oil.

Under an argon atmosphere, 0.1 M solution of Li_2_CuCl_4_ in THF (2.5 mL, 0.25 mmol, 0.1 equiv.) and 2.0 M solution of propylmagnesium bromide (3.75 mL, 7.5 mmol, 3.0 equiv.) were added to a solution of the previously obtained tosylate (2.5 mmol, 1.0 equiv.) in tetrahydrofuran (6 mL) at −20 °C. The mixture was stirred overnight at room temperature. Afterward, the reaction was quenched with aqueous ammonium chloride (10 mL), extracted with ethyl acetate, washed with brine, and dried over anhydrous sodium sulfate. Following concentration, the product was purified using column chromatography on silica gel with hexane to yield the (*S*)-10-methyltetradec-1-ene ((*S*)-**9**) (0.44 g, 83%) as a colorless oil. [α]_D_^25^ = +1.42 (c 1.69, CHCl_3_). ^1^H NMR (500 MHz, CDCl_3_) δ 5.82 (ddt, *J* = 16.9, 10.2, 6.7 Hz, 1H), 5.00 (d, *J* = 17.1 Hz, 1H), 4.93 (d, *J* = 10.2 Hz, 1H), 2.04 (q, *J* = 7.0 Hz, 2H), 1.39–1.36 (m, 3H), 1.28–1.20 (m, 14H), 1.10–1.06 (m, 2H), 0.89 (t, *J* = 6.8 Hz, 3H), 0.84 (d, *J* = 6.6 Hz, 3H). ^13^C NMR (126 MHz, CDCl_3_) δ 139.43, 114.23, 37.25, 36.95, 33.99, 32.89, 30.13, 29.72, 29.51, 29.33, 29.12, 27.23, 23.21, 19.88, 14.33. HRMS (ESI, *m*/*z*): calculated for [M + H]^+^ C_15_H_31_ 211.2420, found: 211.2414.

### 3.9. Synthesis of (S)-10-Methyltetradecan-1-ol ((S)-***10***)

Under an argon atmosphere, 9-borabicyclo [3.3.1] nonane (9-BBN) (12.0 mL, 0.5 M in THF, 6.0 mmol, 3 equiv.) was added to a solution of (*S*)-10-methyltetradec-1-ene ((*S*)-**9**) (0.42 g, 2.0 mmol, 1 equiv.) in dry tetrahydrofuran (16 mL) slowly at room temperature. The mixture was stirred for 12 h, followed by the addition of sodium hydroxide solution (4.0 mL, 3 M, 12.0 mmol, 6 equiv.) at 0 °C. After stirring for 30 min, the temperature was lowered to −20 °C, and hydrogen peroxide solution (4.0 mL, 30%) was added slowly. The reaction was then allowed to proceed for 3 h at room temperature and was quenched with saturated aqueous ammonium chloride (10 mL). The organic layer was separated, and the aqueous layer was extracted with ethyl acetate (3 × 30 mL). The combined organic phases were washed with brine (30 mL), dried over anhydrous sodium sulfate, and concentrated under reduced pressure to yield the crude product. This product was purified by silica gel column chromatography using a petroleum ether/ethyl acetate mixture (5:1) to afford (*S*)-10-methyltetradecan-1-ol ((*S*)-**10**) (0.41 g, 90% yield) as a colorless oil. [α]_D_^25^ = +2.08 (c 0.96, CHCl_3_). ^1^H NMR (500 MHz, CDCl_3_) δ 3.64 (t, *J* = 6.7 Hz, 2H), 1.59–1.54 (m, 2H), 1.48 (s, 1H), 1.35–1.20 (m, 19H), 1.13–1.05 (m, 2H), 0.88 (t, *J* = 6.9 Hz, 3H), 0.83 (d, *J* = 6.6 Hz, 3H). ^13^C NMR (126 MHz, CDCl_3_) δ 63.25, 37.24, 36.92, 32.96, 32.88, 30.15, 29.78, 29.77, 29.59, 29.48, 27.22, 25.89, 23.19, 19.86, 14.31. HRMS (ESI, *m*/*z*): calculated for [M + Na]^+^ C_15_H_32_ONa 251.2345, found: 251.2356.

### 3.10. Synthesis of (S)-14-Methyloctadec-1-ene ((S)-***12***)

Following the procedure previously described for the synthesis of (*S*)-**9**, (*S*)-10-methyltetradecan-1-ol ((*S*)-**10**) (0.30 g, 1.3 mmol, 1.0 equiv.) and 1.0 M solution of but-3-en-1-ylmagnesium bromide (3.9 mL, 3.9 mmol, 3.0 equiv.) were reacted to yield (*S*)-14-methyloctadec-1-ene ((*S*)-**12**) (0.28 g, 80% yield) as a colorless oil. [α]_D_^25^ = +8.44 (c 0.43, CHCl_3_). ^1^H NMR (500 MHz, CDCl_3_) δ 5.82 (ddt, *J* = 16.9, 10.2, 6.7 Hz, 1H), 5.02–4.90 (m, 2H), 2.04 (q, *J* = 6.9 Hz, 2H), 1.40–1.34 (m, 3H), 1.29–1.23 (m, 22H), 1.08 (q, *J* = 8.1 Hz, 2H), 0.90–0.87 (m, 3H), 0.84 (d, *J* = 6.6 Hz, 3H). ^13^C NMR (126 MHz, CDCl_3_) δ 139.44, 114.22, 37.26, 36.94, 33.98, 32.89, 30.19, 29.88, 29.84, 29.83, 29.77, 29.67, 29.50, 29.32, 29.11, 27.24, 23.21, 19.88, 14.33. HRMS (ESI, *m*/*z*): calculated for [M + K]^+^ C_19_H_38_K 305.2605, found: 305.2607.

### 3.11. Synthesis of (R)-10-Methyltetradec-1-ene ((R)-***9***)

Following the procedure previously described for the synthesis of (*S*)-**9**, (*S*)-2-methylundec-10-en-1-ol ((*S*)-**7**) (0.5 g, 2.7 mmol, 1.0 equiv.) and 2.0 M solution of propylmagnesium bromide (4.05 mL, 8.1 mmol, 3.0 equiv.) were reacted to yield (*R*)-10-methyltetradec-1-ene ((*R*)-**9**) (0.48 g, 84% yield) as a colorless oil. [α]_D_^25^ = −0.33 (c 1.20, CHCl_3_). ^1^H NMR (500 MHz, CDCl_3_) δ 5.82 (ddt, *J* = 16.9, 10.2, 6.7 Hz, 1H), 4.99 (dq, *J* = 17.1, 1.6 Hz, 1H), 4.93 (ddt, *J* = 10.2, 2.2, 1.2 Hz, 1H), 2.07–2.02 (m, 2H), 1.39-1.35 (m, 3H), 1.33–1.20 (m, 14H), 1.11–1.05 (m, 2H), 0.89 (t, *J* = 6.9 Hz, 3H), 0.84 (d, *J* = 6.6 Hz, 3H). ^13^C NMR (126 MHz, CDCl_3_) δ 139.43, 114.22, 37.24, 36.94, 33.98, 32.89, 30.12, 29.71, 29.50, 29.33, 29.12, 27.23, 23.21, 19.88, 14.33. HRMS (ESI, *m*/*z*): calculated for [M + H]^+^ C_15_H_31_ 211.2420, found: 211.2415.

### 3.12. Synthesis of (R)-10-Methyltetradecan-1-ol ((R)-***10***)

Following the procedure previously described for the synthesis of (*S*)-**10**, (*R*)-10-methyltetradec-1-ene ((*R*)-**9**) (0.44 g, 2.1 mmol, 1 equiv.) and 9-Borabicyclo(3.3.1)nonane (9-BBN) (12.6 mL, 0.5 M in THF, 6.3 mmol, 3 equiv.) were reacted to yield (*R*)-10-methyltetradecan-1-ol ((*R*)-**10**) (0.42 g, 88% yield) as a colorless oil. [α]_D_^25^ = −0.44 (c 0.92, CHCl_3_). ^1^H NMR (500 MHz, CDCl_3_) δ 3.64 (t, *J* = 6.7 Hz, 2H), 1.59–4.54 (m, 2H), 1.39–1.19 (m, 20H), 1.11–1.05 (m 2H), 0.88 (t, *J* = 6.9 Hz, 3H), 0.83 (d, *J* = 6.6 Hz, 3H). ^13^C NMR (126 MHz, CDCl_3_) δ, 63.25, 37.24, 36.93, 32.96, 32.88, 30.15, 29.79, 29.77, 29.59, 29.49, 27.22, 25.89, 23.19, 19.86, 14.31. HRMS (ESI, *m*/*z*): calculated for [M + Na]^+^ C_15_H_32_ONa 251.2345, found: 251.2363.

### 3.13. Synthesis of (R)-14-Methyloctadec-1-ene ((R)-***12***)

Following the procedure previously described for the synthesis of (*S*)-**9**, (*R*)-10-methyltetradecan-1-ol ((*R*)-**10**) (0.34 g, 1.5 mmol, 1.0 equiv.) and 1.0 M solution of but-3-en-1-ylmagnesium bromide (4.5 mL, 4.5 mmol, 3.0 equiv.) were reacted to yield (*R*)-14-methyloctadec-1-ene ((*R*)-**12**) (0.31 g, 78% yield) as a colorless oil. [α]_D_^25^ = −0.34 (c 1.17, CHCl_3_). ^1^H NMR (500 MHz, CDCl_3_) δ 5.82 (ddt, *J* = 16.9, 10.2, 6.7 Hz, 1H), 4.99 (d, *J* = 17.1 Hz, 1H), 4.93 (d, *J* = 10.2 Hz, 1H), 2.04 (q, *J* = 7.1 Hz, 2H), 1.39–1.36 (m, 3H), 131–1.20 (m, 22H), 1.11–1.06 (m, 2H), 0.89 (t, *J* = 6.9 Hz, 3H), 0.84 (d, *J* = 6.5 Hz, 3H). ^13^C NMR (126 MHz, CDCl_3_) δ 139.43, 114.22, 37.27, 36.95, 33.99, 32.90, 30.20, 29.89, 29.85, 29.84, 29.79, 29.68, 29.51, 29.33, 29.12, 27.26, 23.22, 19.88, 14.33. HRMS (ESI, *m*/*z*): calculated for [M + Na]^+^ C_19_H_38_Na289.2866, found: 289.2877.

### 3.14. Synthesis of (S)-14-Methyloctadecan-2-one ((S)-***1***)

Under an oxygen atmosphere, to a stirred solution of (*S*)-14-methyloctadec-1-ene ((*S*)-**12**) (0.13 g, 0.5 mmol, 1.0 equiv.) in 7:1 DMF: H_2_O (8 mL) was added PdCl_2_ (17.7 mg, 0.1 mmol, 0.2 equiv.) and CuCl_2_ (50.0 mg, 0.5 mmol, 1.0 equiv.) at room temperature. The reaction was stirred overnight at room temperature and filtered off, and the filtrate was treated with a solution of 1N HCl (5 mL). The aqueous solution was extracted with ethyl acetate (3 × 20 mL). The combined organic phases were washed with brine (3 × 30 mL), dried over anhydrous sodium sulfate, and concentrated under reduced pressure to yield the crude product. This product was purified by silica gel column chromatography using a petroleum ether/ethyl acetate mixture (20:1) to afford (*S*)-14-methyloctadecan-2-one ((*S*)-**1**) (0.10 g, 73% yield) as a colorless oil. [α]_D_^25^= +0.92 (c 0.43, CHCl_3_). ^1^H NMR (500 MHz, CDCl_3_) δ 2.41 (t, *J* = 7.5 Hz, 2H), 2.13 (s, 3H), 1.59–1.55 (m, 2H), 1.36–1.32 (m, 1H), 1.27–1.32 (m, 22H), 1.11–1.04 (m, 2H), 0.88 (t, *J* = 6.9 Hz, 3H), 0.83 (d, *J* = 6.6 Hz, 3H). ^13^C NMR (126 MHz, CDCl_3_) δ 209.55, 43.99, 37.25, 36.93, 32.88, 30.17, 30.00, 29.86, 29.80, 29.76, 29.62, 29.55, 29.49, 29.34, 27.23, 24.03, 23.20, 19.87, 14.32. HRMS (ESI, *m*/*z*): calculated for [M + H]^+^ C_19_H_39_O 283.2995, found: 283.2985.

### 3.15. Synthesis of (R)-14-Methyloctadecan-2-one ((R)-***1***)

Following the procedure previously described for the synthesis of (*S*)-**1**, (*R*)-14-methyloctadec-1-ene ((*R*)-**12**) (0.16 g, 0.6 mmol, 1.0 equiv.) and PdCl_2_ (21.3 mg, 0.12 mmol, 0.2 equiv.) and CuCl_2_ (60.0 mg, 0.6 mmol, 1.0 equiv.) were reacted to yield (*R*)-14-methyloctadec-1-ene ((*R*)-**12**) (0.12 g, 70% yield) as a colorless oil. [α]_D_^25^ = −0.89 (c 1.79, CHCl_3_). ^1^H NMR (500 MHz, CDCl_3_) δ 2.41 (t, *J* = 7.5 Hz, 2H), 2.13 (s, 3H), 1.57–1.55 (m, 2H), 1.36–1.32 (m, 1H), 1.28–1.19 (m, 22H), 1.11–1.04 (m, 2H), 0.88 (t, *J* = 6.9 Hz, 3H), 0.83 (d, *J* = 6.6 Hz, 3H). ^13^C NMR (126 MHz, CDCl_3_) δ 209.51, 43.97, 37.24, 36.92, 32.87, 30.17, 29.98, 29.85, 29.80, 29.75, 29.62, 29.55, 29.48, 29.33, 27.23, 24.03, 23.19, 19.86, 14.31. HRMS (ESI, *m*/*z*): calculated for [M + H]^+^ C_19_H_39_O 283.2995, found: 283.2988.

### 3.16. Synthesis of (S)-4-Isopropyl-3-tetradecanoyloxazolidin-2-one ((S)-***14***)

Following the procedure previously described for the synthesis of (*R*)-***5***, tetradecanoic acid (**13**) (1.37 g, 6.0 mmol, 1.0 equiv.) and (*S*)-4-benzyl-2-oxazolidinone (*S*)-**4** (0.93 g, 7.2 mmol, 1.2 equiv.) were reacted to yield (*S*)-4-isopropyl-3-tetradecanoyloxazolidin-2-one ((*S*)-**14**) (1.65 g, 81% yield) as a white solid. The melting point was 56.0–57.0 °C; [α]_D_^25^= +56.03 (c 3.02, CHCl_3_). ^1^H NMR (500 MHz, Chloroform-*d*) δ 4.42 (dt, *J* = 7.1, 3.4 Hz, 1H), 4.25 (t, *J* = 8.7 Hz, 1H), 4.19 (dd, *J* = 9.1, 3.0 Hz, 1H), 3.00–2.94 (m, 1H), 2.87–2.81 (m, 1H), 2.41–2.32 (m, 1H), 1.68-1.61 (m, 2H), 1.35–1.24 (m, 20H), 0.90 (d, *J* = 7.1 Hz, 3H), 0.87 (t, *J* = 6.7 Hz, 6H). ^13^C NMR (126 MHz, CDCl_3_) δ 173.56, 154.19, 63.42, 58.50, 35.65, 32.04, 29.80, 29.77, 29.73, 29.61, 29.50, 29.48, 29.26, 28.51, 24.60, 22.81, 18.09, 14.78, 14.24. HRMS (ESI, *m*/*z*): calculated for [M + H]^+^ C_20_H_38_O_3_N 340.2846, found: 340.2838.

### 3.17. Synthesis of (S)-4-Isopropyl-3-((S)-2-methyltetradecanoyl)oxazolidin-2-one ((S, S)-***15***)

Following the procedure previously described for the synthesis of (*R*, *R*)-**6**, (*S*)-4-isopropyl-3-tetradecanoyloxazolidin-2-one ((*S*)-**14**) (1.36 g, 4.0 mmol, 1.0 equiv.) and methyl iodide (1.25 mL, 20.0 mmol, 5.0 equiv.) were reacted to yield (*S*)-4-isopropyl-3-((*S*)-2-methyltetradecanoyl)oxazolidin-2-one ((*S*, *S*)-**15**) (1.1 g, 75% yield) as a colorless oil. [α]_D_^25^ = +60.61 (c 1.96, CHCl_3_). ^1^H NMR (500 MHz, CDCl_3_) δ 4.46–4.43 (m, 1H), 4.25 (t, *J* = 8.7 Hz, 1H), 4.19 (dd, *J* = 9.1, 2.9 Hz, 1H), 3.75–3.68 (m, 1H), 2.38–2.32 (m, 1H), 1.74–1.67 (m, 1H), 1.37–1.32 (m, 1H), 1.30–1.24 (m, 20H), 1.19 (d, *J* = 6.9 Hz, 3H), 0.91 (d, *J* = 7.0 Hz, 3H), 0.89–0.86 (m, 6H). ^13^C NMR (126 MHz, CDCl_3_) δ 177.46, 153.81, 63.32, 58.57, 37.86, 33.26, 32.06, 29.81, 29.81, 29.78, 29.73, 29.65, 29.49, 28.57, 27.45, 22.82, 18.08, 17.99, 14.82, 14.25. HRMS (ESI, *m*/*z*): calculated for [M + H]^+^ C_21_H_40_O_3_N 354.3003, found: 354.2996.

### 3.18. Synthesis of (S)-2-Methyltetradecan-1-ol ((S)-***16***)

Following the procedure previously described for the synthesis of (*R*)-**7**, (*S*)-4-isopropyl-3-((*S*)-2-methyltetradecanoyl)oxazolidin-2-one ((*S*, *S*)-**15**) (0.7 g, 2 mmol, 1.0 equiv.) and LiAlH_4_ (0.27 g, 7.0 mmol, 3.5 equiv.) were reacted to yield (*S*)-2-methyltetradecan-1-ol ((*S*)-**16**) (0.35 g, 77% yield, ≥99% ee, determined by ^1^H NMR analysis of the ester derived from (*S*)-MTPACl) as a colorless oil. [α]_D_^25^= −10.59 (c 1.96, CHCl_3_). ^1^H NMR (500 MHz, CDCl_3_) δ 3.51 (dd, *J* = 10.5, 5.8 Hz, 1H), 3.41 (dd, *J* = 10.5, 6.6 Hz, 1H), 1.65–1.56 (m, 1H), 1.42–1.35 (m, 2H), 1.33–1.26 (m, 20H), 1.13–1.07 (m, 1H), 0.91 (d, *J* = 6.7 Hz, 3H), 0.88 (t, *J* = 7.0 Hz, 3H). ^13^C NMR (126 MHz, CDCl_3_) δ 68.58, 35.92, 33.30, 32.07, 30.10, 29.83, 29.82, 29.80, 29.51, 27.13, 22.84, 16.73, 14.26. HRMS (ESI, *m*/*z*): calculated for [M + Na]^+^ C_15_H_32_ONa 251.2345, found: 251.2365.

### 3.19. Synthesis of (R)-4-Isopropyl-3-tetradecanoyloxazolidin-2-one ((R)-***14***)

Following the procedure previously described for the synthesis of (*R*)-***5***, tetradecanoic acid (**13**) (1.60 g, 7.0 mmol, 1.0 equiv.) and (*R*)-4-benzyl-2-oxazolidinone (*R*)-**4** (1.08 g, 8.4 mmol, 1.2 equiv.) were reacted to yield (*R*)-4-isopropyl-3-tetradecanoyloxazolidin-2-one ((*S*)-**14**) (2.04 g, 86% yield) as a white solid. The melting point was 57.0–58.0 °C; [α]_D_^25^ = −57.20 (c 2.0, CHCl_3_). ^1^H NMR (500 MHz, Chloroform-*d*) δ 4.43 (dt, *J* = 8.1, 3.4 Hz, 1H), 4.25 (t, *J* = 8.7 Hz, 1H), 4.19 (dd, *J* = 9.1, 3.0 Hz, 1H), 3.00–2.94 (m, 1H), 2.87–2.81 (m, 1H), 2.40–2.34 (m, 1H), 1.69–1.60 (m, 2H), 1.36–1.25 (m, 20H), 0.91 (d, *J* = 7.0 Hz, 3H), 0.87 (t, *J* = 6.8 Hz, 6H). ^13^C NMR (126 MHz, CDCl_3_) δ 173.58, 154.20, 63.43, 58.51, 35.66, 32.05, 29.81, 29.78, 29.74, 29.62, 29.51, 29.49, 29.27, 28.52, 24.61, 22.82, 18.10, 14.79, 14.25. HRMS (ESI, *m*/*z*): calculated for [M + H]^+^ C_20_H_38_O_3_N 340.2846, found: 340.2837.

### 3.20. Synthesis of (R)-4-Isopropyl-3-((R)-2-methyltetradecanoyl)oxazolidin-2-one ((R, R)-***15***)

Following the procedure previously described for the synthesis of (*R*, *R*)-**6**, (*R*)-4-isopropyl-3-tetradecanoyloxazolidin-2-one ((*R*)-**14**) (1.53 g, 4.5 mmol, 1.0 equiv.) and methyl iodide (1.40 mL, 22.5 mmol, 5.0 equiv.) were reacted to yield (*R*)-4-isopropyl-3-((*R*)-2-methyltetradecanoyl)oxazolidin-2-one ((*R*, *R*)-**15**) (1.2 g, 74% yield) as a colorless oil. [α]_D_^25^ = −72.8 (c 1.87, CHCl_3_). ^1^H NMR (500 MHz, CDCl_3_) δ 4.46–4.43 (m, 1H), 4.26 (t, *J* = 8.7 Hz, 1H), 4.19 (dd, *J* = 9.1, 3.0 Hz, 1H), 3.75–3.68 (h, *J* = 6.9 Hz, 1H), 2.38–2.31 (m, 1H), 1.73–1.67 (m, 1H), 1.37–1.34 (m, 1H), 1.31–1.24 (m, 20H), 1.19 (d, *J* = 6.9 Hz, 3H), 0.91 (d, *J* = 7.0 Hz, 3H), 0.89–0.85 (m, 6H). ^13^C NMR (126 MHz, CDCl_3_) δ 177.47, 153.81, 63.32, 58.58, 37.87, 33.27, 32.06, 29.82, 29.81, 29.78, 29.74, 29.66, 29.49, 28.58, 27.45, 22.83, 18.09, 18.00, 14.83, 14.26. HRMS (ESI, *m*/*z*): calculated for [M + H]^+^ C_21_H_40_O_3_N 354.3003, found: 354.2995.

### 3.21. Synthesis of (R)-2-Methyltetradecan-1-ol ((R)-***16***)

Following the procedure previously described for the synthesis of (*R*)-**7**, (*R*)-4-isopropyl-3-((*R*)-2-methyltetradecanoyl)oxazolidin-2-one ((*R*, *R*)-**15**) (0.78 g, 2.2 mmol, 1.0 equiv.) and LiAlH_4_ (0.29 g, 7.7 mmol, 3.5 equiv.) were reacted to yield (*R*)-2-methyltetradecan-1-ol ((*R*)-**16**) (0.40 g, 80% yield, ≥99% ee, determined by ^1^H NMR analysis of the ester derived from (*S*)-MTPACl) as a colorless oil. [α]_D_^25^ = +6.50 (c 1.35, CHCl_3_). ^1^H NMR (500 MHz, CDCl_3_) δ 3.51 (dd, *J* = 10.5, 5.7 Hz, 1H), 3.41 (dd, *J* = 10.5, 6.6 Hz, 1H), 1.64–1.57 (m, 1H), 1.39–1.35 (m, 2H), 1.33–1.26 (m, 20H), 1.15–1.09 (m, 1H), 0.91 (d, *J* = 6.7 Hz, 3H), 0.88 (t, *J* = 6.9 Hz, 3H). ^13^C NMR (126 MHz, CDCl_3_) δ 68.59, 35.92, 33.30, 32.07, 30.10, 29.83, 29.82, 29.80, 29.51, 27.13, 22.84, 16.73, 14.26. HRMS (ESI, *m*/*z*): calculated for [M + Na]^+^ C_15_H_32_ONa 251.2345, found: 251.2358.

### 3.22. Synthesis of (S)-6-Methyloctadec-1-ene ((S)-***17***)

Following the procedure previously described for the synthesis of (*S*)-**9**, (*S*)-2-methyltetradecan-1-ol ((*S*)-**16**) (0.25 g, 1.1 mmol, 1.0 equiv.) and 1.0 M solution of but-3-en-1-ylmagnesium bromide (3.3 mL, 3.3 mmol, 3.0 equiv.) were reacted to yield (*S*)-6-methyloctadec-1-ene ((*S*)-**17**) (0.21 g, 71% yield) as a colorless oil. [α]_D_^25^ = −0.25 (c 1.59, CHCl_3_). ^1^H NMR (500 MHz, CDCl_3_) δ 5.86–5.78 (m, 1H), 5.02–5.01 (m, 1H), 4.94–4.92 (m, 1H), 2.07–2.00 (m, 2H), 1.41–1.36 (m, 2H), 1.31–1.26 (m, 23H), 1.13–1.06 (m, 2H), 0.90–0.87 (m, 3H), 0.85 (d, *J* = 6.6 Hz, 3H). ^13^C NMR (126 MHz, CDCl_3_) δ 139.45, 114.24, 37.20, 36.72, 34.31, 32.82, 32.10, 30.18, 29.89, 29.87, 29.83, 29.53, 27.24, 26.59, 22.86, 19.84, 14.28. HRMS (ESI, *m*/*z*): calculated for [M + Na]^+^ C_19_H_38_ Na 289.2866, found: 289.2874.

### 3.23. Synthesis of (R)-6-Methyloctadec-1-ene ((R)-***17***)

Following the procedure previously described for the synthesis of (*S*)-**9**, (*R*)-2-methyltetradecan-1-ol ((*R*)-**16**) (0.23 g, 1.0 mmol, 1.0 equiv.) and 1.0 M solution of but-3-en-1-ylmagnesium bromide (3.0 mL, 3.0 mmol, 3.0 equiv.) were reacted to yield (*S*)-6-methyloctadec-1-ene ((*S*)-**17**) (0.19 g, 70% yield) as a colorless oil. [α]_D_^25^ = +1.11 (c 0.72, CHCl_3_). ^1^H NMR (500 MHz, CDCl_3_) δ 5.86–5.78 (m, 1H), 5.01–4.98 (m, 1H), 4.94–4.92 (m, 1H), 2.05–2.00 (m, 2H), 1.41–1.36 (m, 2H), 1.29–1.26 (m, 23H), 1.13–1.26 (m, 2H), 0.89–0.86 (m, 3H), 0.85 (d, *J* = 6.6 Hz, 3H). ^13^C NMR (126 MHz, CDCl_3_) δ 139.46, 114.24, 37.20, 36.71, 34.31, 32.81, 32.09, 30.17, 29.88, 29.86, 29.82, 29.52, 27.23, 26.59, 22.85, 19.84, 14.28. HRMS (ESI, *m*/*z*): calculated for [M + Na]^+^ C_19_H_38_ Na 289.2866, found: 289.2875.

### 3.24. (S)-6-Methyloctadecan-2-one ((S)-***2***)

Following the procedure previously described for the synthesis of (*S*)-**1**, ((*S*)-6-methyloctadec-1-ene ((*S*)-**17**) (0.13 g, 0.5 mmol, 1.0 equiv.), PdCl_2_ (17.7 mg, 0.10 mmol, 0.2 equiv.) and CuCl_2_ (50.0 mg, 0.5 mmol, 1.0 equiv.) were reacted to yield (*S*)-6-methyloctadecan-2-one ((*S*)-**2**) (96.1mg, 68% yield) as a colorless oil. [α]_D_^25^ = −0.93 (c 1.72, CHCl_3_). ^1^H NMR (500 MHz, CDCl_3_) δ 2.41–2.38 (m, 2H), 2.13 (s, 3H), 1.64–1.49 (m, 2H), 1.37–1.33 (m, 1H), 1.30–1.23 (m, 22H), 1.12–1.05 (m, 2H), 0.87 (t, *J* = 7.0 Hz, 3H), 0.85 (d, *J* = 6.6 Hz, 3H). ^13^C NMR (126 MHz, CDCl_3_) δ 209.49, 44.28, 37.03, 36.66, 32.79, 32.07, 30.13, 29.99, 29.86, 29.84, 29.80, 29.51, 27.17, 22.84, 21.57, 19.68, 14.26. HRMS (ESI, *m*/*z*): calculated for [M + H]^+^ C_19_H_39_O 283.2995, found: 283.2985.

### 3.25. (R)-6-Methyloctadecan-2-one ((R)-***2***)

Following the procedure previously described for the synthesis of (*S*)-**1**, ((*R*)-6-methyloctadec-1-ene ((*R*)-**17**) (0.11 g, 0.4 mmol, 1.0 equiv.), PdCl_2_ (14.2 mg, 0.08 mmol, 0.2 equiv.) and CuCl_2_ (40.0 mg, 0.4 mmol, 1.0 equiv.) were reacted to yield (*R*)-6-methyloctadecan-2-one ((*R*)-**2**) (76.8 mg, 66% yield) as a colorless oil. [α]_D_^25^ = +0.88 (c 0.45, CHCl_3_).^1^H NMR (500 MHz, CDCl_3_) δ 2.40 (t, *J* = 7.3 Hz, 2H), 2.13 (s, 3H), 1.65–1.51 (m, 2H), 1.37–1.35 (m, 1H), 1.30–1.26 (m, 22H), 1.12–1.05 (m, 2H), 0.89 (t, *J* = 6.7 Hz, 3H), 0.85 (d, *J* = 6.6 Hz, 3H). ^13^C NMR (126 MHz, CDCl_3_) δ 209.55, 44.30, 37.04, 36.67, 32.80, 32.08, 30.14, 30.01, 29.87, 29.85, 29.81, 29.51, 27.18, 22.85, 21.59, 19.70, 14.27. HRMS (ESI, *m*/*z*): calculated for [M + H]^+^ C_19_H_39_O 283.2995, found: 283.2983.

## 4. Conclusions

To summarize, a concise and novel method to synthesize the active sex pheromone components of *L. dharma dharma* ((*S*)-**1** and (*S*)-**2**) and their enantiomers was developed. We successfully synthesized the target sex pheromone component (*S*)-**1** in seven steps with a total yield of 29%, and the target sex pheromone component (*S*)-**2** in five steps with a total yield of 23%, and their optical purities were both greater than 99%. Compared with reported synthetic strategies, our synthetic route has the advantages of cheap starting materials, short route, high overall synthesis yield, and excellent optical purity of the target compound. Our strategy centrally employed Evans’ chiral auxiliary to construct the stereocenter, while the pheromone skeleton was built efficiently through Grignard cross-coupling reactions, hydroboration–oxidation, and Wacker oxidation. The synthetic sex pheromone components will advance the study of communication mechanisms in lichen moths, species identification, and supported ecological management.

## Data Availability

The data presented in this article are available in the Appendix A.

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
