# Peer review of "Enantioselective Synthesis of the Active Sex Pheromone Components of the Female Lichen Moth, Lyclene dharma dharma, and Their Enantiomers"

_molecules, 2024, doi:10.3390/molecules29122918_

Round 1

Reviewer 1 Report

Comments and Suggestions for Authors

The authors describe a relatively straightforward chemical synthesis of two molecules which are female-produced sex pheromones of the lichen moth, L dharma dharma. These two molecules are straight-chain (with 1 methyl branch) ketones eighteen carbons in length, and for the key step to install the methyl group asymmetrically, Evans’ chiral auxiliary is used. The authors claim 99.5 % enantiospecificity (see note about this in my comments below). The chemistry is straightforward and the yields are good, and my only issues with this manuscript are mainly editorial ones (see comments below). There is an issue if the tense of the verbs, which varies between past and present for much of the manuscript. I would suggest using all past tense for consistency’s sake.

Editorial comments:

-line 13, recommend “has been” instead of “have been” (tense of the verb).

-line 27, recommend the comma after 14-methyl-2-octadecanone, then a space.

-line 31, recommend “and” after “(Figure 1),”

-line 34, recommend “was previously based” instead of “was mainly based”

-line 34, insert a space after “enantiomers”

-line 34, recommend “Mori completed” not “Mori has completed”.

-line 42, insert “an” before “alkyl lithium”.

-line 47, recommend “is still in demand” not “was still demanding”. Also, recommend “was” instead of “is” in other instances in the manuscript, such as on lines 58 and 91.

-line 50, insert “a” before “Grignard”

-line 51, recommend “yielded” not “yielding”

-line 65, recommend “after” instead of “including”

-line 70 insert “was” after sequence. Likewise, line 83, insert “which” before “also”, line 140 insert “which” before “afforded”

-line 72 “of” instead of “with”

-line 76, can the NMR really distinguish the minor diastereomer if it is only present in 0.5 %? If this is true, no elaboration is really needed. Just my thoughts…

-line 91, “alkene” not “alkenes”

-line 92, capitalize the M of Methyl, and “to give” not “gave”

-line 97, “a” instead of “an”

-line 98, “smoothly produced” not “produced smoothly”

-line 99, omit “the”

-Scheme 3, the way it is drawn looks like the Grignard reagents 8 and 11 are introduced to 7 or 10 before the tosylation occurs, when it is really the other way around. I would recommend putting the Grignard reagents 8 and 11 under the arrow with the Li2CuCl4. Likewise in Scheme 6. Also, in the Scheme 3 caption, I would recommend a space after (S)-

-line 107, I would recommend “gave” not “to give” (tense of the verb).

-line 108, I would recommend “as” not “at”

-line 137 “shown” not “showed”

-line 144, the specific rotation that the authors obtained seems to be the right sign, but of a significantly different magnitude from the literature, is there any explanation for this? A different solvent, perhaps? An explanation would be helpful.

-line 158 the authors list “Beijing China” for all of the instruments that they used, are all the instrument manufacturers actually based in Beijing? Is this where the instruments came from, or where the measurements were taken?

-lines 164 and 188, insert a hyphen after (undec-10-enoyl)

-line 164, 5 should be in bold.

-regarding the experimental NMR data, for compound 5 (both R and S) one 13C resonance seems to be missing; is this just because two resonances in the alkyl chain, or possibly the two methyls on the chiral auxiliary, have the same chemical shift? If this is the case, no explanation should be necessary, just as long as the data is correct. Likewise, compounds S- and R-14, (S, S)- and (R, R)-15 (missing one C13 resonance). Also, two 13C resonances seem to be missing from S- and R-16, likewise S- and R-17, and S- and R-2. If this is just a case of multiple resonances occurring at the same chemical shift, no explanation should be necessary.

-However, for compound S-1, when I total up the hydrogen resonances in the proton NMR data I get 46, not 38 (should be 38 hydrogens in compound 1). Please correct this data.

-line 233, and elsewhere the experimental data write-up, I would definitely recommend “were reacted to give” not “were reacted to” This occurs multiple places in the experimental, and should be corrected.

-line 267, the HRMS data found definitely is a typo (shouldn’t be 181898). Please correct.

-line 280, insert “were” before “added”, likewise on line 294.

-line 284, what solvent system was used for the column chromatography?

-line 293, remove one of the spaces before 9-BBN.

-for compounds S-and R-12, and R-9, the tosylate was formed before the cuprate coupling, correct? This could be mentioned to avoid confusion. Likewise S- and R-17.

-line 359, “The aqueous solution was extracted” not “Aqueous solution extracted”

-line 419, “13” should be bold.

-lines 477 and 487, omit “and”

-line 499, “was” instead of “were”

-lines 501-502, recommend “…total yield of 23 %, and their optical purities were both greater…” as an alternative wording.

-line 505 “…components will advance the study…” not “…components would be advance the study…”

-line 560, remove the periods before and after “alpha”

-line 567, give initials for the author Kitching

-line 569, maybe “with unprecedented” not just “unprecedented”?

Line 572, “Ei” is not an author’s initials, please correct.

Comments on the Quality of English Language

No real issues, besides several typos and editorial comments; just that the authors should use past tense consistently throughout the manuscript and not lapse into present tense in certain parts, like in the experimental. I have pointed these out in my comments to the authors.

Reviewer 2 Report

Comments and Suggestions for Authors

This article describes a highly enantioselective synthesis of two pheromones. Although the compounds have been synthesized before, the synthesis is novel and uses established synthetic steps, so it is likely to be easily adopted by others to synthesize the target compounds. The work is thorough and well structured. Several minor issues can be resolved as indicated in the attached pdf file.

The discussion should include a comparison with the previous synthesis both in overall yield and in steps. The high DE of the Evnas alkylation (99.5:0.5) should be supported by previous publications and discussed.

At least for the final products, a standard error calculation of the [a] values would be helpful.

The biggest problem is that in at least five cases the HR-MS data of the enantiomers match exactly. Even the error of a missing number and point is repeated in the enantiomer. This means that the data have been fabricated, which compromises the whole manuscript, which the readers read trusting its honesty. Therefore, I cannot recommend publication, because other data may have been fabricated as well.

Comments on the Quality of English Language

The English has to be improved in soem occasions noted in the pdf file.

Reviewer 3 Report

Comments and Suggestions for Authors

The manuscript “Enantioselective Synthesis of the Active Sex Pheromone Components of the Female Lichen Moth, L. dharma dharma and Their Enantiomers” is devoted to the enantioselective synthesis of (S)-14-methyloctadecan-2-one ((S)-1), (S)-6-methyloctadecan-2-one ((S)-2) and their enantiomers. The products are active sex pheromones and can be useful in ecological biotechnology.

In my opinion, this manuscript suits to the scope of Molecules. I recommend that it can be accepted after major revision.

Some more comments for the authors:

1.      SI with NMRs must be added.

Round 2

Reviewer 2 Report

Comments and Suggestions for Authors

I appreciate that the original spectra of the HR-MS data were included in the author response. In this time with more and more fabricated data we should avoid any suspicion of fabrication.

Reviewer 3 Report

Comments and Suggestions for Authors

The authors provided required changes. The manuscript is ready to be accepted.